# Potential European Geographical Distribution of *Gnathotrichus materiarius* (Fitch, 1858) (Coleoptera: Scolytinae) under Current and Future Climate Conditions

Radosław Witkowski [1,*], Marcin K. Dyderski [2], Marta Bełka [1] and Andrzej Mazur [1]

1 Faculty of Forestry and Wood Technology, Department of Forest Entomology and Pathology, Poznań University of Life Sciences, 60-625 Poznań, Poland; marta.belka@up.poznan.pl (M.B.); andrzej.mazur@up.poznan.pl (A.M.)
2 Institute of Dendrology, Polish Academy of Sciences, Parkowa 5, 62-035 Kórnik, Poland; mdyderski@man.poznan.pl
* Correspondence: radoslaw.witkowski@up.poznan.pl

**Abstract:** *Gnathotrichus materiarius* (Fitch, 1858) is an alien ambrosia beetle from North America, that has been spreading across Europe since the 1930s. The species infests coniferous trees, excavating galleries in sapwood. However, to date very few studies have predicted changes in ambrosia beetle habitat suitability under changing climate conditions. To fill that gap in the current knowledge, we used the MaxEnt algorithm to estimate areas potentially suitable for this species in Europe, both under current climate conditions and those forecasted for the years 2050 and 2070. Our analyses showed areas where the species has not been reported, though the climatic conditions are suitable. Models for the forecasted conditions predicted an increase in suitable habitats. Due to the wide range of host trees, the species is likely to spread through the Balkans, the Black Sea and Caucasus region, Baltic countries, the Scandinavian Peninsula, and Ukraine. As a technical pest of coniferous sapwood, it can cause financial losses due to deterioration in quality of timber harvested in such regions. Our results will be helpful for the development of a climate-change-integrated management strategy to mitigate potential adverse effects.

**Keywords:** ambrosia beetle; bark beetle; MaxEnt; insect pest; alien species; niche modelling; biological invasions





## 1. Introduction

Biological invasions are among the most important phenomena affecting not only biodiversity, ecosystem functioning, but also the economy [1–3]. Invasive species can lead to shifts in species composition, affecting the course of succession, as well as modifying nutrient cycling, carbon sequestration, and water balance in ecosystems [4]. In terms of the impact on economy in the last 50 years, invasive species have been responsible globally for financial losses of over 1.288 trillion USD [5].

Bark beetles and ambrosia beetles are arboreal organisms, and are represented in large numbers among alien species due to introduction events in numerous regions [6,7]. These insects are associated with wood, where they develop. Microsites under the bark of round wood and wood products protect insects against unfavourable conditions during transport. Therefore, these insects can survive long journeys and become introduced far from their natural ranges [8]. Global trade development, leading to increases in the quantity and speed of long-distance transport, has been the main cause of accidental introductions of bark and ambrosia beetles worldwide [6,9–11], classified as a contaminant pathway [12]. Europe is a target area of bark and ambrosia beetle introductions, mainly from Asia and North America [10,13].

Ambrosia beetles (Coleoptera, Scolytinae, and Platypodinae) are a polyphyletic group covering numerous species from the Scolytinae subfamily, mainly Xyleborini and Xyloterini tribes, and from the Platypodinae subfamily [14]. Ambrosia beetles differ from bark beetles in foraging type: bark beetles feed on phloem, while ambrosia beetles feed on fungi growing in tunnels created within a host plant [15–17]. The wide range of host plants and the ability to carry fungal spores in their mycangia (unique structures on the insects' bodies adapted for the transport of symbiotic fungi) are life-history traits that are particularly important for the success of their naturalisation and spread. Furthermore, the possibility of asexual reproduction of Xyleborini, allowing individual females to produce offspring outside a natural range where their population was previously absent or very sparse and scattered, is a crucial aspect in promoting the spread of this group [14]. Alien species from the Scolytinae subfamily naturalized in Europe represent over 12% of all European Scolytinae, of which most are ambrosia beetles [6,10,13]. These species usually infect weakened or dead trees and rarely kill healthy plants [18]. However, fungi carried in their mycangia might be pathogenic to particular host species, leading to economic losses [19–23]. *Gnathotrichus materiarius* (Fitch, 1858) (Figure 1) is one of the alien ambrosia beetles spreading across Europe [11,24,25]. In Europe, it was recorded for the first time in France in the 1930s [26]. It has since spread eastward and currently occurs in Austria [27,28], Belgium [29], Czechia [30], Finland [31], Spain [32], Netherlands [33], Germany [34], Poland [35], Slovenia [36], Switzerland [24], Sweden [37], Great Britain [11], and Italy [38]. The species was also reported during phytosanitary inspections of products imported to New Zealand [39]. *Gnathotrichus materiarius* (Coleoptera, Scolytinae) is an ambrosia beetle from the Corthylini tribe [27]. It is a monogamous species, with a sex ratio of 1:1, with no sibling mating nor asexual reproduction, which distinguishes *G. materiarius* from the Xyleborini tribe species [25,40–42]. Larvae and imagines of *G. materiarius* feed on a symbiotic fungus *Ambrosiozyma monospora* (Saito) van der Walt (1972) (syn. *Endomycopsis fasciculata* Batra) [43–45], recorded both in the native and secondary range of *G. materiarius* [43,46]. Imagines of the studied species occur during the whole growing period, with flight culmination in May or June [38,45,47]. *Gnathotrichus materiarius* is a technical pest of coniferous wood in the USA and Europe [38,48,49]. It prefers *Pinus*, while also infesting numerous genera of conifers: *Abies, Larix, Picea, Pinus, Pseudotsuga, Thuja*, or *Tsuga* [25,33,50]. Due to excavation galleries in the lower part of the trunk, it is of economic importance as a pest, decreasing the technical quality and economic value of affected timber [25,51,52]. Moreover, fungi growing in the tunnels can cause wood staining [25,31], although this phenomenon is not very common. So far, *G. materiarius* has not caused significant losses to forest management in areas beyond its natural range, as evidenced mainly by the relatively limited attention devoted to this species in scientific studies of these regions.

The progression rate of climate change intensifies its impact on ecosystems [31,52–56]. One of the predicted impacts of climate change on forest ecosystems is the increasing frequency and intensity of insect outbreaks, including alien species [57]. Due to their short life cycles and their strict dependence on temperature, insects are exceptionally responsive to climate change [58,59]. Moreover, increasing temperatures and drought intensity lead to physiological stress in trees, decreasing their resistance to bark and ambrosia beetle infestation [60–63]. The poor overall condition and lack of vigour among host plants favour the naturalisation and spread of alien species and increases their population size [18].

Previously published models predicting the future distribution of Scolytinae under changing climate conditions were created primarily for phloeophagous bark beetles of high economic importance [64–69]. In contrast, among alien ambrosia beetle species occurring in Europe, such models have been developed only for *Xylosandrus compactus* and *X. crassiusculus* [70]. For other parts of the world, data on their distributions are scarce [70,71]. Due to information deficiency and differences in the biology and ecology of bark and ambrosia beetles [49], the need to develop species distribution models is urgent and increasingly pressing.

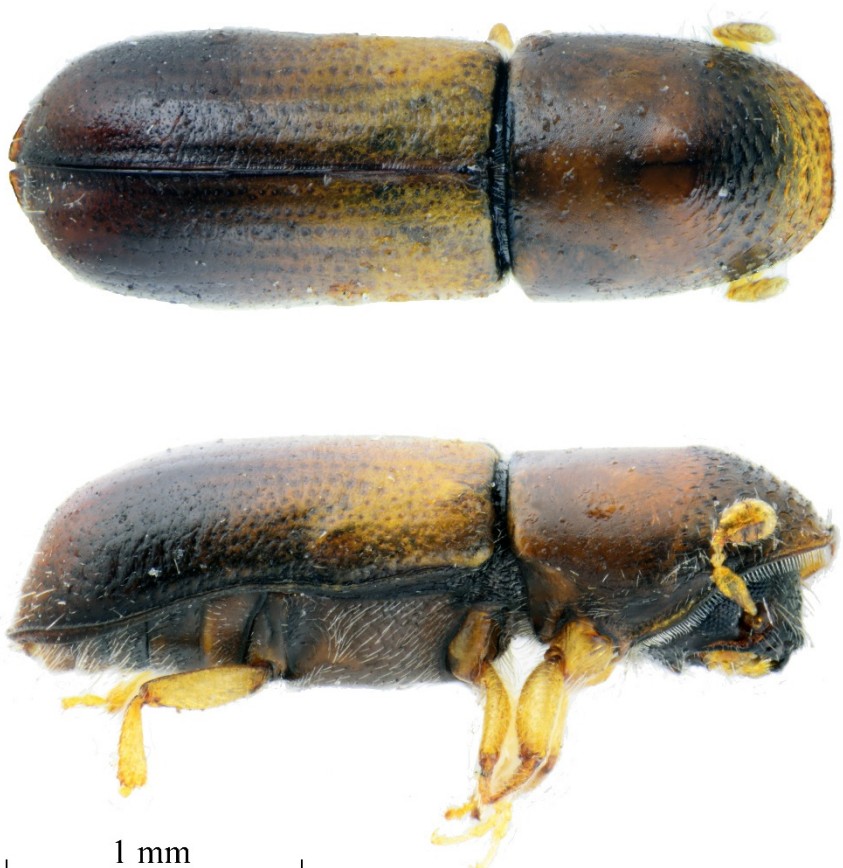

**Figure 1.** Dorsal and lateral view of an adult male *Gnathotrichus materiarius* (R. Witkowski).

*Gnathotrichus materiarius* differs from two *Xylosandrus* spp. for which distribution models have already been developed. *Xylosandrus compactus* and *X. crassiusculus* infest broadleaved as well as coniferous tree species, and have different sex determination systems. *Gnathotrichus materiarius* is a diploidal species, while *Xylosandrus* spp. are haplodiploid inbreeders, with beetles sib-mating before the new generation of females emerges from the gallery system [40–42]. Xyleoborini males do not cover long distances and sometimes do not even leave galleries [40]. The mating behaviour of outbreeding *G. materiairus* with a balanced sex ratio is different. Males and females leave galleries and spread by active flight to find both a mate and substrate to infest [41]. Thus, *G. materiarius* is an ideal candidate for a model species, exemplifying trends for other invasive ambrosia beetles [72–74], and our results can serve as a proxy for other species sharing similar life-history traits, either already present or potentially appearing in Europe in the future. Therefore, we aimed to develop a *G. materiarius* distribution model to assess which areas will be suitable for its spread under various climate change scenarios. We hypothesized that (1) the current climatic conditions support further spread of *G. materiarius* in Europe (unsaturated climatic niche), and (2) under climate change scenarios for the 2050s and 2070s, the area of climatic suitability for *G. materiarius* (climatically suitable area) will expand.

## 2. Materials and Methods

### 2.1. Data Collection

We compiled all available records of *G. materiarius* from public databases [47,75,76] containing data from peer-reviewed publications [77–91], peer-reviewed publications not included in datasets [11,25,35,92,93], and previously unpublished communications included in a PhD dissertation [94], as along with observations collected by the authors of this paper between 2017 and 2020, (Supplementary Table S1). After data compilation,

we excluded observations with incorrect coordinates or metadata suggesting a mismatch between locality description and coordinates. After excluding duplicated coordinates, we obtained 1448 data points. To reduce over-sampling in some regions and under-sampling in others, we randomly selected only one occurrence from each 0.25° grid raster cell [95]. This stratified downsampling prevented overestimation of species occurrence in regions with higher sampling effort and decreased spatial autocorrelation of data. As a result, we obtained 807 single observations that were used for analysis.

We downloaded a dataset of 19 bioclimatic variables from the WorldClim 2.1 database [96]. These variables, derived from monthly temperature and precipitation records, are widely used when modelling species distributions [97–99]. To avoid including highly correlated predictors, we checked the Pearson correlation coefficients for the pairs of bioclimatic variables. We removed the variables that were most strongly correlated with others, assuming r > |0.7| as the threshold value. The final set of variables used for model development included six bioclimatic variables (Table 1).

**Table 1.** Bioclimatic variables used in the study.

| Abbreviation | Parameter |
| --- | --- |
| BIO2 | Mean Diurnal Range (Mean of monthly (max–temp–min temp)) (°C) |
| BIO7 | Temperature Annual Range (BIO5-BIO6) (°C) |
| BIO11 | Mean Temperature of Coldest Quarter (°C) |
| BIO15 | Precipitation Seasonality (Coefficient of Variation: mean/SD × 100) (%) |
| BIO16 | Precipitation of Wettest Quarter (mm) |
| BIO17 | Precipitation of Driest Quarter (mm) |

We used future climate projects provided by the IPCC 6th Assessment Report, based on Shared Socioeconomic Pathways (SSPs) [100]. These scenarios reflect uncertainties in possible trajectories of climate change mitigation. We chose four scenarios available in the WorldClim 2.1 database [96]: SSP126 (sustainability, the most optimistic scenario reflecting RCP2.6 from the fifth report), SSP245 (middle of the road, moderate scenario reflecting RCP4.5), SSP370 (regional rivalry, not used in the fifth report) and SSP585 (fossil-fuel based development or business-as-usual, reflecting RCP8.5). We used each SSP outcomes for four global circulation models (GCMs): IPSL-CM6A-LR (France), MRI-ESM2-0 (Japan), CanESM5 (Canada), and BCC-CSM2-MR (China), representing half of eight available GCMs for all the four SSPs. We prepared predictions for two timelines: 2041–2060 and 2061–2080. We decided to use these timelines as they are the most common frameworks for species distribution models [101–103].

*2.2. Modelling Species Distribution*

Before analyses, we divided the datasets into a training set (80% of observations), used for model development, and an independent validation set (20% of observations). The use of an independent dataset for model evaluation prevents model overfitting, which could limit our conclusions to the dataset range. Due to the presence-only character of distributional data, we used the MaxEnt algorithm to develop species distribution models. MaxEnt has been developed to process presence-only data [104,105]; in contrast to parametric models and other classification tools, it does not need absence data in the theoretical assumptions, instead using background data (the so-called pseudoabsences). We used default MaxEnt settings. For the species, we randomly selected 10,000 pseudoabsence points (background points). Thus, MaxEnt analysed patterns of presence distinct from the background data. Therefore, the prevalence of background points makes the model more conservative, requiring a stronger signal than would be the case for equal proportions of presences and pseudoabsences [105]. We assessed model quality using the area under the receiver operator curve (AUC) as a metric depending on true positive and true negative rates (i.e., positive and negative rates overlapping the real and predicted occurrence). The output of the MaxEnt model is the probability that a particular species can occur in a partic-

ular raster cell. To obtain presence–absence information, we calculated the threshold—the probability value with the highest sum of sensitivity (true positive rate) and specificity (true negative rate). Such an approach balances false negatives and false positives [106]. The MaxEnt model also provides information about variable importance, expressed as per cent contribution to the model, as well as response curves, showing how the model output changes along with studied variables. We used the "dismo" package [107] for MaxEnt model development, and the "raster" [108] and "sf" [109] packages for spatial data processing. According to the model, we calculated potential range saturation as a proportion of sampled points and the number of pixels suitable for particular species occurrence.

We applied models to maps of current and future climatic scenarios to obtain predictions of *G. materiarius* distribution. For each SSP we averaged predicted species occurrence probability across the four GCMs, to reduce uncertainty connected with particular GCMs [110–112]. Then, maps with threshold values (true/false) were used to estimate the changes in the potential range, changing the values on maps with the future potential range from 1 to 2 [113]. As a result of the calculations presented below, four different variants of change in the range of the species were estimated: (i) areas still unsuitable for colonisation ($0 − 2 \times 0 = 0$)—no changes, the species was not present, and the prevailing conditions will remain unsuitable for colonisation; (ii) range expansion ($0 − 2 \times 1 = −2$)—areas potentially suitable for colonisation; (iii) range contraction ($1 − 2 \times 0 = 1$)—areas where the species is currently present but will fall outside the future optimal climate; (iv) persistence ($1 − 2 \times 1 = −1$)—areas where the species is currently present and will remain within the optimal climate [113].

## 3. Results

### 3.1. G. materiarius Distribution Model

The Maxent models produced reliable *G. materiarius* distributions with a very high AUC value (0.98). The predicted threshold of presence was 0.2548. Precipitation of the driest quarter (bio17) was the most important predictor, with an average contribution of 58.7%, whilst less critical were the annual temperature range (bio7) and the precipitation seasonality (bio15) (Figure 2). Individual response curves of the two strongest bioclimatic variables showed that the predicted probability of the presence of *G. materiarius* was positively correlated with these factors (Figure 2).

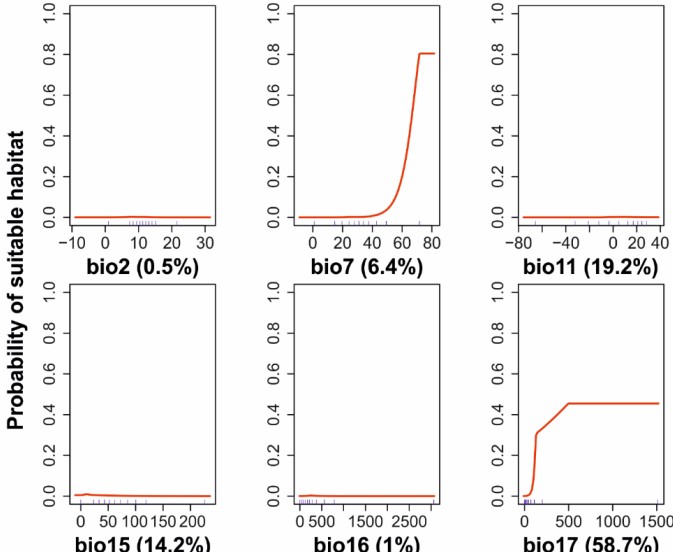

**Figure 2.** Response curves present how predicted niche suitability for *Gnathotrichus materiarius* changed along particular predictor gradients. Numbers in parentheses indicate the relative importance of variables. Variable importance, i.e., the proportion of variation, explained by particular variable based on AUC [113] gain for a single feature.

### 3.2. Current Potential Distribution

The MaxEnt model developed for current climate conditions reported habitat suitability for 98% of localities where *G. materiarius* was already documented. Furthermore, it identified areas where *G. materiarius* was not noted, but climate conditions allow for its development. In Europe, such highly suitable areas are concentrated in the Balkans (Albania, Bulgaria, Bosnia and Herzegovina, Montenegro, Romania, Serbia, Slovenia), the Black Sea and Caucasus region (Armenia, Georgia, Turkey, Russia), the Baltic countries (Lithuania, Latvia, and Estonia), the Scandinavian Peninsula and Ukraine (Figure 3). Our model indicated that currently *G. materiarius* can find suitable climate conditions across 13.1% of Europe.

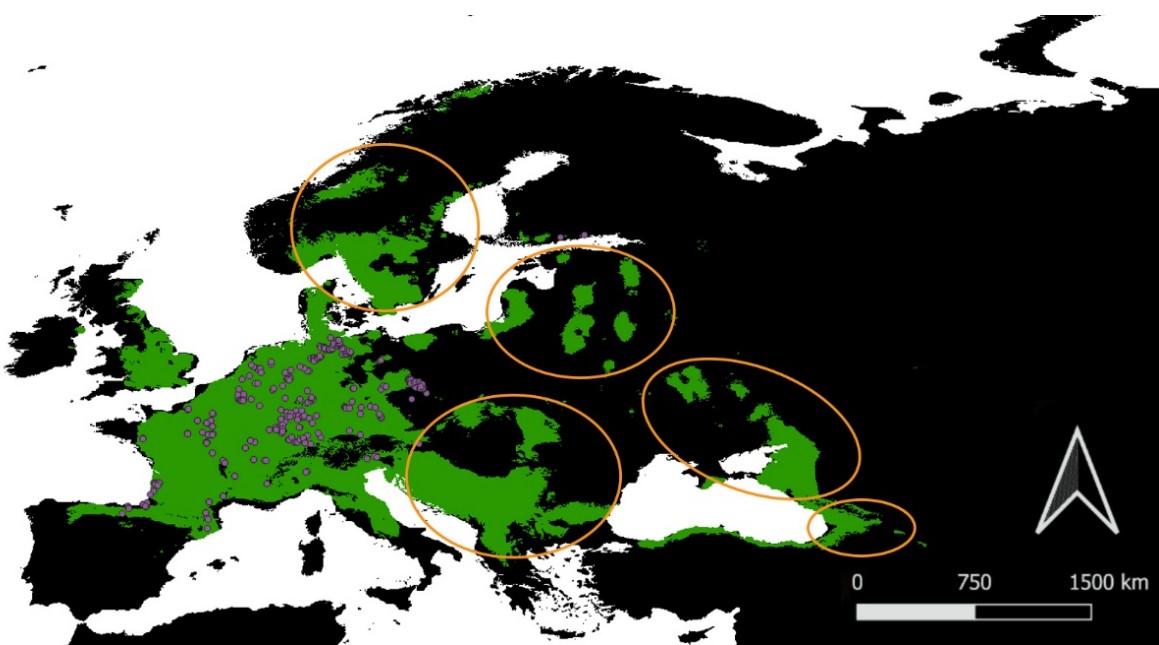

**Figure 3.** Predicted habitat suitability for *Gnathotrichus materiarius* under current climate conditions (green area), localities of known occurrence records (violet points), and crucial new potential sites that have not been colonised yet.

### 3.3. Predicted Range Shifts

The model predicted an increase in habitat suitability for *G. materiarius* in European countries for all the scenarios in both periods (Figure 4.). For each subsequent SSP, from the most optimistic (SSP126) to the least (SSP585), an increase was obtained in the number of cells where *G. materiarius* can find suitable conditions (Table 2). In the period 2041–2060 predicted habitat suitability increases to approx. 20.4%–25.2%, while in the period 2041–2060 it increases to ca. 20.8%–31.2% of the area in Europe. Predictions for the most pessimistic SSP585 in the period 2061–2080 indicate that with changing climate, the range of the species will increase mainly eastward to the Ural Mountains and northward almost to the Arctic Circle on the Scandinavian peninsula. Only slight shifts in the western and southern range of the species were forecasted.

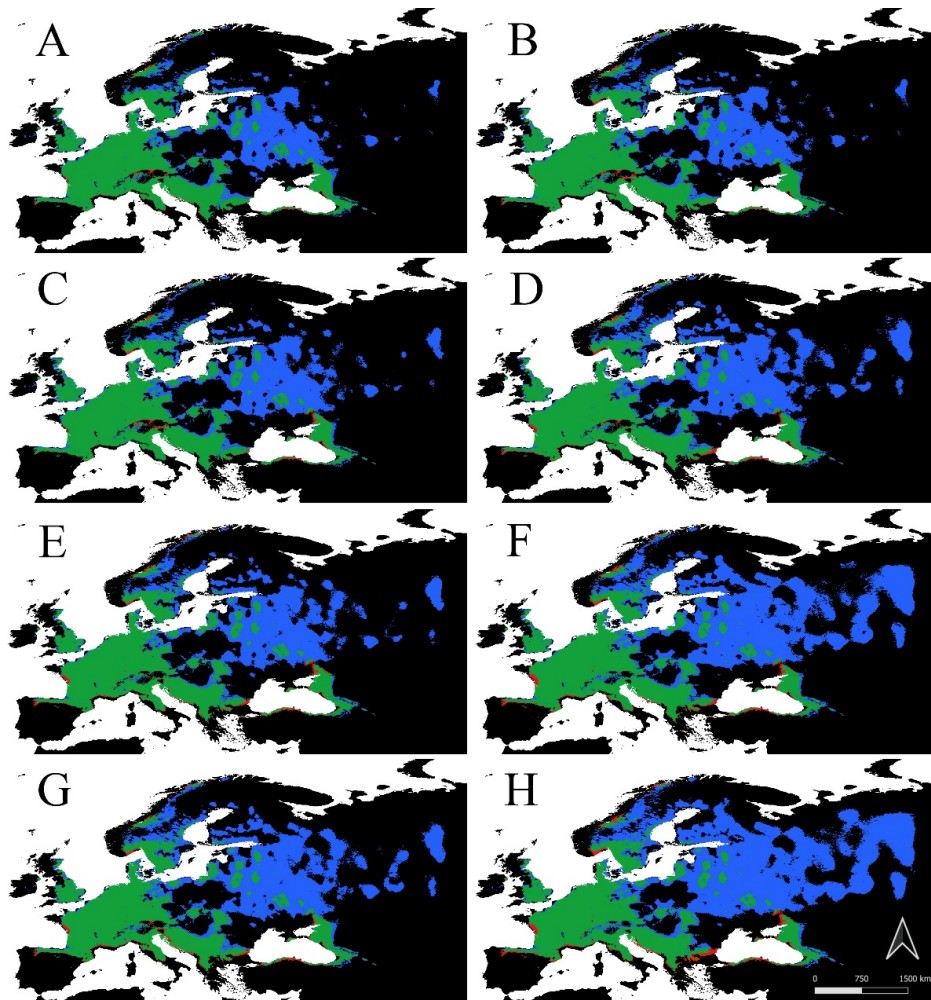

**Figure 4.** Maps of predicted shifts of *Gnathotrichus materiarius* climatic suitability, averaged across four GCMs in Europe for SSP126 (**A**,**E**) SSP245 (**B**,**F**) SSP370 (**C**,**G**) SSP585 (**D**,**H**) in two timelines: 2041–2060 (**A**–**D**), 2061–2080 (**E**–**H**). Green—persistence, blue—range expansion, red—range contraction.

**Table 2.** Predicted shifts of *Gnathotrichus materiarius* range size under analyzed SSP and timelines related to actual habitat suitability.

| Shared Socioeconomic Pathway (SSP) and Timeline | Range Expansion (Absent/Present) (%) | Persistence (Present/Present) (%) | Range Contraction (Present/Absent) (%) | Net Shift (Expansion–Contraction) |
|---|---|---|---|---|
| SSP126 2041–2060 | 7.53 | 11.35 | 0.25 | 7.28 |
| SSP245 2041–2060 | 9.09 | 11.27 | 0.32 | 8.77 |
| SSP370 2041–2060 | 10.39 | 11.15 | 0.45 | 9.94 |
| SSP585 2041–2060 | 12.53 | 11.13 | 0.47 | 12.06 |
| SSP126 2061–2080 | 8.04 | 11.29 | 0.31 | 7.73 |
| SSP245 2061–2080 | 11.88 | 11.14 | 0.46 | 11.43 |
| SSP370 2061–2080 | 16.27 | 11.05 | 0.55 | 15.72 |
| SSP585 2061–2080 | 18.86 | 10.87 | 0.72 | 18.14 |

## 4. Discussion and Conclusions

Our results indicate that the species has not yet already reached its maximum extent within the climatically suitable area. The study indicates that the species can spread and reach more sites, because a low level of niche saturation is typical of the beginning of the second stage of invasion—the 'log phase', which comes after the 'lag phase' [114]. Our

prediction developed for current climate conditions (Figure 1) pointed out the most likely areas where the species can spread. The Balkans and the Baltic countries are highly suitable regions situated a short distance from the known locations of *G. materiarius*, and there are no natural barriers that would prevent the colonisation of this area. Although the Black Sea region is a considerable distance from these places, the natural spread of *G. materiarius* from the Balkans along the coastline to the Caucasus cannot be excluded. It should be remembered that trade plays an essential role in the spread of species, and was the probable source of the appearance of *G. materiarius* in Europe. Another factor that supports our prediction is a wide range of host tree species for *G. materiarius*, which would not limit the colonization process [115].

Climate change has a natural effect on the distribution of species and can also affect some aspects of their biology. Shortening the development cycle of beetles and thus increasing the number of generations during the year may have serious economic effects, as has already been observed in the case of other bark and ambrosia beetles [58,116]. In south-western Poland, where the species has been present since 2015 [35], it causes economic losses as a technical pest. However, additional studies are necessary to determine the exact scale of the phenomenon. Changes in precipitation patterns can significantly impact trees, triggering stress-induced ethanol accumulation [117]. Therefore, higher susceptibility to ambrosia beetle infestations and thus an increase in their economic importance may be reliably inferred.

We have sparse occurrence data including coordinates for most alien bark and ambrosia beetle species present in Europe. The distribution of *G. materiarius* is one of the better-documented, and has allowed us to develop a high-accuracy model. Developing such models is economically crucial because results can be used to design more detailed surveys in future, and thus facilitate better planning for the usage of limited funds and human resources. Furthermore, such models can be used to estimate the rate and direction of invasions of ambrosia beetle species with similar ecological habitats, but with less well recognised distributions.

However, it should be taken into account that climate change affects species and their populations directly (temperature, precipitation) and indirectly (by affecting antagonistic and symbiotic organisms or food bases) [118,119]. Therefore the effects of climate change are difficult to predict precisely. Moreover, the studied species has not colonised all suitable habitats within the study area. Consequently, it is not in equilibrium, which also affects the reliability of estimations for the studied species. In order to improve the model it will be necessary to collect coordinates of the places of occurrence, and make these available in the databases. Additionally, especially in high-risk areas, monitoring with dedicated or wide-range lures [41,120] is indicated for the rapid identification of threats, which may be crucial to ensure their reduction or control.

Our study provided a species distribution model based on climatic variables and presence-only data. Due to its correlative nature, our model explains patterns in data and does not reflect physiological processes [105,106]. Moreover, our model allows only conclusions for the predicted climatic niche, not actual occurrences that would be affected by other factors, especially dispersal limitation and land-use patterns.

The presence of a suitable host plant is also essential for beetle development. Models for *Pinus sylvestris* and *Picea abies* predict shifts to the north in their ranges, with potential contraction in the south [118]. However, it is hard to anticipate which species will replace them. So far, such models have been developed only for a subset of native and a few alien tree species, neglecting those with relatively small ranges. However, such species can become widespread and increase in economic importance [121]. The climate is the primary determinant of species distributions on a large spatial scale, affecting other interactions [122]. Another source of uncertainty in our model is the dependence on GCMs used in the study [110–112]. We decreased this uncertainty by averaging model predictions for four different GCMs.

Nevertheless, the results obtained in the study indicate which regions are more vulnerable to the occurrence of *G. materiarius* (Figure 4.), and thus the area where phytosanitary services need to focus to this species. Such actions could slow down the spread of the species in Europe through transported wood and wood products, which have been the major source of alien bark and ambrosia beetle invasions [123]. Phytosanitary measures including heating and fumigation may reduce the probability of invasions, although due to the depth of the galleries, which may cover the entire sapwood, they do not guarantee the neutralisation of all specimens. Another threat is related to ornamental plants, mainly Thuja and Tsuga, which are transported alive, thus limiting the range of the methods used for treating raw wood (e.g., heating and drying).

Our study provides the first assessment of *G. materiarius* potential current and future ranges in Europe. This study, together with models obtained for two *Xylosandrus* species, are the only predictions of the potential spread of ambrosia beetles. Therefore, our study provides quantitative foundations for spread prevention and risk assessments necessary for conservation biogeography [124].

**Supplementary Materials:** The following supporting information can be downloaded at: https://www.mdpi.com/article/10.3390/f13071097/s1, Table S1: Coordinates used to develop the model of the occurrence of *Gnathotrichus materiarius*.

**Author Contributions:** Concept of the study: R.W., A.M. and M.B. Data collection: R.W. Data analysis methodology: M.K.D. Data analysis: M.K.D., R.W. All authors participated in manuscript preparation: A.M., M.B., M.K.D., R.W. All authors have read and agreed to the published version of the manuscript.

**Funding:** Publication was co-financed within the framework of the Polish Ministry of Science and Higher Education's program: "Regional Excellence Initiative" in the years 2019–2022, project no. 005/RID/2018/19, financing amount 1,200,000,000 PLN. The study was partially supported by the Institute of Dendrology, Polish Academy of Sciences, Kórnik, Poland.

**Institutional Review Board Statement:** Not applicable.

**Informed Consent Statement:** Not applicable.

**Data Availability Statement:** All data are included in the text and the Supplementary Material.

**Acknowledgments:** The authors would like to thank anonymous reviewers for their valuable comments and remarks and employees of the Polish State Forests for their help in field work.

**Conflicts of Interest:** The authors declare that they have no competing interests.

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
