# Peer review of "Potential European Geographical Distribution of Gnathotrichus materiarius (Fitch, 1858) (Coleoptera: Scolytinae) under Current and Future Climate Conditions"

_forests, doi:10.3390/f13071097_

Round 1

Reviewer 1 Report

The author predicts the distribution model of an invasive forest pest in Europe. Which is good and important to forestry in Europe. but I would not agree to accept it. Because the whole paper did not write like a scientific paper and many language mistakes. The author did not carefully write and check it before submission. It wastes reviewers' time.

In the introduction. I would highly suggest the author to point out the subject Gnathotrichus materiarius earlier in the introduction. Hiding it in the end part of the Introduction make the explanation of biology and important of this species become indirect. Besides, as an invasive pest, most people wonder how it damages the forest or tree in other areas. this is important data and the reason for this study. Please provide it.

Line 89 more whole "species studied" to introduction. It did not explain the materials and methods here. the author should describe where those location data come from. The known data and history should be in the introduction. 

Line96 "biocontrol"? control the product? 

Line 98 it's clear this beetle prefer pine.

Line 109 In the appendix, there are only 965 points, how it become 1448 after excluding the duplicated points? also the important 807 single observations should be provided.

line 133 2041-2060 are too far from now. most people concerned the result from now to the next 20 years

Line 136-137 why divided the dataset? explain here.

line 161. where is the asterisk?

Figure 2 and 3  both figure are missing the national boundary. Which make reading the accurate location or countries very difficult. 

Figure 2 several violet points did not under the green area. which should be explained.

i understand this paper focus on Europe. but people would like to see the predicted distribution of whole world. Author should present the world map in the main result or in the supplemental material.

Appendix 1

1- Please add full name of the country in the table. 

2- "Atkinson 2018" is not the original citation. In Atkinson's website. Most data were cited from  Wood's book. please cite the original citation. The same to GBIF2021

3- replace "original data" with "in this study"

Author Response

Thank you for devoting your time to our manuscript. We are thankful for constructive comments, however there are not so many of them to say that our manuscript does not present scientific values.

  1. Reviewer: "In the Introduction. I would highly suggest the author to point out the subject Gnathotrichus materiarius earlier in the Introduction. Hiding it in the end part of the Introduction make the explanation of biology and important of this species become indirect.
    Reply: Thank you for this comment. We agree that the structure of the Introduction should be modified to present the object of research earlier. We have moved one of the paragraphs to meet this remark.
  1. Reviewer: Besides, as an invasive pest, most people wonder how it damages the forest or tree in other areas. this is important data and the reason for this study. Please provide it."
    Reply: Unfortunately, there is little data on this issue. Currently, we are witnessing the importance of the species, but we do not have enough materials to support them at the moment. We are aware that this issue is important, therefore we conduct observations aimed at presenting the importance of the species in forestry. Nevertheless, at the moment, the brief information in the Discussion, "In south-western Poland, where the species has been present since 2015 [50], it causes economic losses as a technical pest. However, additional studies are necessary to determine the exact scale of the phenomenon." must be sufficient.
  1. Reviewer: "Line 89 more whole "species studied" to introduction. It did not explain the materials and methods here. "
    Reply: Paragraf moved to Introduction as suggested.
  1. Reviewer: Line96 "biocontrol"? control the product?"
    Reply: Agree, it was an incorrect use of the word. Thank you for catching up. The text was corrected.
  1. Reviewer: "Line 98 it's clear this beetle prefer pine."
    Reply: Information added.
  1. Reviewer: "Line 109 In the appendix, there are only 965 points, how it become 1448 after excluding the duplicated points? also the important 807 single observations should be provided."
    Reply: Previously, the wrong file was attached. The relevant file is already in the system.
  1. Reviewer: "line 133 2041-2060 are too far from now. most people concerned the result from now to the next 20 years"
    Reply: We cannot agree with this statement as most papers focusing on species distribution models aim to show predictions for even further future - common timelines are 2041-2060 and 2061-2080. Moreover, 2041 is literally next 20 years. We added an explanation to the text: "We used these timelines as they are the most common frameworks in species distribution models [Anibaba et al. 2022; Polaina et al. 202; Seidl 2018]".

Anibaba, Q. A., Dyderski, M. K., & Jagodziński, A. M. Predicted range shifts of invasive giant hogweed (Heracleum mantegazzianum) in Europe. Sci. Total Environ., 825, 154053, https://doi.org/10.1016/j.scitotenv.2022.154053 (2022).

Polaina, E., Soultan, A., Pärt, T., & Recio, M. R. The future of invasive terrestrial vertebrates in Europe under climate and land-use change. Environ. Res. Lett. 16(4), 044004, https://doi.org/10.5061/dryad.kkwh70s37 (2021).

Seidl, R., Klonner, G., Rammer, W., Essl, F., Moreno, A., Neumann, M., & Dullinger, S. Invasive alien pests threaten the carbon stored in Europe's forests. Nat. Commun. 9(1), 1-10, https://doi.org/10.1038/s41467-018-04096-w (2018).

  1. Reviewer: Line 136-137 why divided the dataset? explain here.
    Reply: In data science, we usually split the dataset into training and validation set. We added a short explanation: "Using an independent dataset for model evaluation prevents model overfitting, which could limit our conclusions to dataset range."

  1. Reviewer: "line 161. where is the asterisk?"
    Reply: We agree that it is misleading. In Table, we showed only these variables we used in the analysis. We deleted the unnecessary sentence.
  1. Reviewer: "Figure 2 and 3  both figure are missing the national boundary. Which make reading the accurate location or countries very difficult."
    Reply: This issue is a matter of taste. Not adding borders was a deliberate decision, and we would like to leave it as it is.
  1. Reviewer: "Figure 2 and 3  both figure are missing the national boundary. Which make reading the accurate location or countries very difficult. "Figure 2 several violet points did not under the green area. which should be explained."
    Reply: It is easy to explain – for model development, we used 80% of observations, however earlier we resampled them in grid to avoid bias towards areas with high sampling effort. High number of observations in W Poland is connected with the high intensity of studies conducted by the Authors.

  1. Reviewer: "i understand this paper focus on Europe. but people would like to see the predicted distribution of whole world. Author should present the world map in the main result or in the supplemental material."
    Reply: We agree that it would be more interesting, however there are no sufficient data coverage of studied species distribution in world scale. Therefore, to maintain minimal coverage, we decided to focus on Europe only. We hope in future to expand our study due to the increasing sampling intensity in many countries.

  1. Reviewer: “Please add full name of the country in the table.”
    Reply: We did it as suggested.

  1. Reviewer: "Atkinson 2018" is not the original citation. In Atkinson's website. Most data were cited from  Wood's book. please cite the original citation. The same to GBIF2021".
    Reply: We solved this "issue" by adding original sources included in datasets. We hope that this meets the Reviewer's expectations.

Reviewer 2 Report

The manuscript presented for review concerns the issue of uropean potential geographical distribution of Gnathotrichus materiarius (Fitch, 1858) (Coleoptera: Scolytinae) under current and future climate. As the authors write, the already published models in which future distribution under changing climate for Scolytinae have been predicted were created primarily for high economic importance cambiofagous bark beetles. In contrast, among alien ambrosia beetle species occurring in Europe, such models have been developed only for Xylosandrus compactus and X. crassiusculus. For other parts of the world, data on their distributions are scarce 65. Due to information deficiency and differences in the biology and ecology of bark and ambrosia beetles, developing species distribution models is urgent. I agree with the authors that the subject discussed in the manuscript requires extensive research and complementation of scientific knowledge. The introduction is written legibly, fully and clearly. Materials and methods: I propose to add photos of the tested species. The methods (successive stages) could be presented in the form of a flowchart - it would be more readable for the recipient than continuous text. The results are presented correctly. The discussion begins with a summary and conclusions - I suggest moving lines 224-241 further down in this chapter.

Author Response

Dear Reviewer,

Thank you for devoting your time to our manuscript. We are thankful for constructive comments.

  1. Reviewer: “I propose to add photos of the tested species.”
    Reply: Figure added to the Introduction.
  1. Reviewer: “The methods (successive stages) could be presented in the form of a flowchart - it would be more readable for the recipient than continuous text.”
    Reply: We agree that the flowchart could be more readable. Unfortunately, it is hard to present this path graphically clearly. We will try to implement it in another publication of a similar nature.
  2. Reviewer: “The discussion begins with a summary and conclusions - I suggest moving lines 224-241 further down in this chapter.”
    Reply: Text moved.

Reviewer 3 Report

The study deals with actual topic and brings interesting results. Text is well arranged and understandable. Modelling of changes in biological systems including changes in species distribution is usually limited by strong simplification of the model (only climatic data are considered) and it is necessary to be aware of this problem. Authors are conscious regarding these issues and some limitations of the model are discussed in the manuscript. However, I suggest also discussing the limitation of G. materiarius future spread arising from distribution of suitable host tree species. Further, it is not clear to me, how the model was adjusted to reflect real climatic requirements of the species. I expect it was based on the presence/absence data from its current range as associated with local climatic characteristics. This should be explicitly stated in the methods. Using of the current introduced range and its climatic specification for model setting should be discussed as another possible limitation of the model outcomes, because current range of the species in Europe might be still restricted by other factors than climate (historical and dispersal limitations, host tree distribution, missing data …).

Some miner comments are mentioned directly in the attached commented manuscript.

Author Response

Dear Reviewer,

Thank you for devoting your time to our manuscript. We are thankful for constructive comments.

  1. Reviewer: “However, I suggest also discussing the limitation of G. materiarius future spread arising from distribution of suitable host tree species.”
    Reply: To meet this suggestion, we added a short paragraph:
    “The presence of a suitable host plant is also essential for developing beetles. Models for Pinus sylvestris and Picea abies, predict shifting their ranges to the north with potential contraction in the southern part [107]. However, it is hard to anticipate which species will be introduced in their place if the models come true because some action will most definitely be taken. So far, such models have not been developed for native tree species with a relatively small population and therefore with little importance, but they may be widespread for economic purposes.”

  2. Reviewer: “Further, it is not clear to me, how the model was adjusted to reflect real climatic requirements of the species. I expect it was based on the presence/absence data from its current range as associated with local climatic characteristics. This should be explicitly stated in the methods.”
    Reply: We already stated that we used a method developed for presence-only data “MaxEnt has been developed to process presence-only data [91,92]”. As it is impossible to obtain true absence data in large spatial scale, species distribution models base on pseudoabsences, that we have also already explained “Therefore, the prevalence of background points makes the model more conservative, as it requires a stronger signal than would be the case for equal proportions of presences and pseudoabsences”.

  1. Reviewer: Using of the current introduced range and its climatic specification for model setting should be discussed as another possible limitation of the model outcomes, because current range of the species in Europe might be still restricted by other factors than climate (historical and dispersal limitations, host tree distribution, missing data …)
    Reply: We tried to explain it in these words:
    “Our study provided a species distribution model based on climatic variables and presence-only data. Due to its correlative nature, our model explains patterns in data and does not reflect physiological processes [9093,9194]. Moreover, our model allows for concluding only about the predicted climatic niche, not actual occurrences, affected by other factors, especially dispersal limitation and land-use patterns.
    The presence of a suitable host plant is also essential for developing beetles. Models for Pinus sylvestris and Picea abies, predict shifting their ranges to the north with potential contraction in the southern part [107]. The presence of a suitable host plant is also essential for developing beetles. Models for Pinus sylvestris and Picea abies, predict shifting their ranges to the north with potential contraction in the southern part [105]. However, it is hard to anticipate which species will replace them. So far, such models have been developed only for a subset of native and few alien tree species, neglecting those with a relatively small ranges. However, such species can become widespread and increase in economic importance [108].
    When accounting for a large spatial scale, the climate is the primary determinant of species distributions, affecting other interactions [108110]. Another source of uncertainty in our model is the dependence on GCMs used in the study [9699,98100,109111]. We de-creased this uncertainty by averaging model predictions for four different GCMs.
    Nevertheless, the results obtained in the study determine which regions are more vulnerable to the occurrence of G. materiarius (Fig 4.)...”
    It is hard to predict all aspects with so many unknowns. History showed that unintentional introductions (with wood and wood materials) in the case of insect dispersion are the overriding factor over natural barriers in the process of their spreading. Therefore, we have limited it to information that has already been included in the manuscript.

  1. Reviewer: “Specify the exact meaning of percentage share (what variable is used ... R2???, decrease in model deviance?)”
    Reply: Variable importance is the proportion of variation explained by particular variable, based on AUC gain for a single feature (Phillips 2006). We added to the caption information 'Variable importance, i.e. the proportion of variation explained by particular variable based on AUC gain for a single feature (Phillips 2006)'
  1. Reviewer: "Some miner comments are mentioned directly in the attached commented manuscript."
    Reply: We have addressed all other of them directly in the manuscript.
